# Synthesis of the Bipyridine-Type Ligand 3-(2-Pyridyl)-5,6-diphenyl-1,2,4-triazine and Structural Elucidation of Its Cu(I) and Ag(I) Complexes

**Antonios Hatzidimitriou, Antonios Stamatiou** 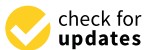**, Dimitrios Tzimopoulos * and Pericles D. Akrivos**

Department of Chemistry, Aristotle University of Thessaloniki, P.O. Box 135, GR-541 24 Thessaloniki, Greece
* Correspondence: dim_tzimopoulos@yahoo.gr

**Abstract:** The synthesis of a substituted diimine with a bipydirine-type backbone, (3-(2-pyridyl)-5,6-diphenyl-1,2,4-triazine, L) and its coordination towards Cu(I) and Ag(I) is studied in the presence of diphosphine ligand bis(diphenylphosphino)methane, dppm. The metal complexes are characterized by IR, $^1$H, and $^{13}$C NMR and single crystal X ray diffraction studies. They are dinuclear, as they are held by diphosphine bridges between the tetrahedral metal centers, forming eight-membered ring with the participation of the bridging diphosphinomethane ligands. Within each ring, the planar orientations of $M_2P_2$ and of all four P atoms are realized. Solid state excitation spectra are dominated by metal-to-ligand charge transfer bands (MLCT), while geometry relaxation permits only low-intensity emission for the copper compound.

**Keywords:** 3-(2-pyridyl)-5,6-diphenyl-1,2,4-triazine; bipydirine; diphosphine; Cu(I); Ag(I)

## 1. Introduction

The α-diimines rank among the most studied ligands in coordination chemistry, due to the chelate effect stabilizing the transition metal compounds [1], and in this respect, they have been tested for a wide variety of applications with a broad spectrum of metals. Complexes of Ag(I) [2] and Cu(I) [3] have used extensively as bacteriostatics and, in some cases, as antitumor agents [4], while silver diphoshpine compounds are reported to possess antitumor activity [5]. Cu(I) compounds with a variety of coordinated ligands have been extensively studied for application in photocatalysis [6] as a major contribution to modern organic synthesis [7], as well as dye-sensitized solar cells [8]. The complexes are considered as state of the art and financially favorable noble metal replacements in light harvesting assemblies. Photosensitizers based on Cu(I) diimine complexes with applications extending to produce oxygen species capable of destroying microorganisms have also been reported [9]. Transition metal diimine complexes generally possess long-lived MLCT excited states, which may be used for several purposes; however, possible geometric modifications upon excitation have been shown to provoke the subsequent quenching of the photoemission process [10]. The problem may be circumvented by introducing a variety of backbone modifications or substitutions on the diimine ligands, aiming at the extension of the excited state lifetime [11]. Along a different line of approach, the introduction to the Cu(I) centre of a biphosphine, especially one with a large bite angle, may also act as relaxation extender [12,13]. Furthermore, such heteroleptic complexes have been proven to offer, besides tunable emission spectra, higher emission quantum yields, as well as adjustable redox properties [14]. The above are facilitated by the concomitant bonding effects of phosphorus atoms as both σ donors and π acceptors to the metal centre. However, most of the heteroleptic Cu(I) complexes with diimines and diphosphines suffer from instability in the solution and a tendency to interact with their environment or undergo internal ligand exchange, quenching their emission properties in the process [15]. Less information is available on the photochemical and photophysical properties of Ag(I)

complexes for which the emission processes are not yet fully understood [16]. However, relative to their Cu(I) counterparts, the Ag(I) complexes reveal more ligand-centered charge transfer processes and weak MLCT excitations [17]. The study of the coordination compounds of bisdiimines has attracted interest in the computational chemistry regime, as well in an effort to compare experimental results and computational predictions for the further utilization of computations for the directed synthetic strategy [18]. Efforts are directed mainly at the tuning of the excited singlet–triplet energy difference, which, if sufficiently small, allows for the repopulation of the singlet state from the triplet one, giving rise to indirect enhance of fluorescence and leading to more efficient light emitting devices [19]. In the present study, we report on the synthesis of a diimine ligand with a substituted triazine ring attached to a typical pyridine one and on the synthesis and structural determination of its complexes with Cu(I) and Ag(I) in the presence of bridging bis(diphenylphosphino)methane.

## 2. Experimental

### 2.1. Materials and Methods of Study

The chemical reagents were used as received, without any further purification. The phosphines and metal salts were products of Merck (Lebanon, NJ, USA), and the amines and aldehydes were for Aldrich (St. Louis, MO, USA). The solvents used were obtained from Aldrich and were of pro-analysis purity. FTIR spectra were recorded in KBr pellets on a FT Perkin-Elmer model 16PC spectrometer (Shelton, CT, USA) in the range 430–4000 cm$^{-1}$. $^1$H and $^{13}$C NMR spectra were recorded on a Bruker Avance DPX 400.13 MHz spectrometer (Billerica, MA, USA) and reported relative to internal TMS standard. Diffuse reflectance spectra were measured on a Jasco V750 UV-Vis absorption spectrophotometer (Easton, MD, USA) equipped with an integrating sphere accessory. Emission and excitation spectra were measured on an Edinburgh Instruments FS5 spectrofluorometer (Livingston, Scotland) equipped with a Xe arc lamp and a red-sensitive Hamamatsu R980 photomultiplier tube (PMT), which is stabilized by cooling. The data were processed by the Fluoracle software provided by the manufacturer.

### 2.2. Synthesis of the Diimine Ligand

Synthetic processes for the ligand, which are currently commercially available, range from the initial typical Diels–Alder reaction [20] to microwave-assisted ones [21], with high yields. Our variation, while maybe more time consuming, however, avoids use of excess refluxing and large volumes of solvent and is described below.

A total of 1 mmol of 2-cyanopyridine was dissolved in 30 mL of isopropanol, and the solution was cooled to 0 °C. The addition of 3 mmol of hydrazine was followed by stirring for 1 h, and then the solution was left to reach room temperature under continuous stirring. Vacuum filtration and washing with 2 × 5 mL hexane produced a yellow product. A total of 1 mmol of the product (2-pyridylhydrazine amide) was dissolved in 30 mL of isopropanol, and 1 mmol of diphenylethanedione was added to it. The mixture was heated to 70 °C and was kept under stirring for 16 h, before being left to reach room temperature. The product was filtered and washed with isopropanol and diethylether, yield 90%.

### 2.3. Synthesis of the Complexes

In a small amount of MeCN was dissolved 1 mmol of the appropriate metal salt (CuBF$_4$ or AgNO$_3$, respectively) and an equimolar amount of the diphoshpine bis(diphenylphospine)methane, dppm. The mixture was allowed to stir at 25 °C for about 45 min. To the resulting turpid solution, a solution of 1 mmol of the ligand L in 10 mL of CH$_2$Cl$_2$ was added. The appearance of an orange or yellow color indicates the immediate formation of a coordination compound. The solution was stirred at room temperature for a further 45 min and then filtrated. To the filtrate, 5 mL of Et$_2$O was added, and the resulting solution was left to evaporate slowly at room temperature. Crystals of the compounds were formed by the third day of evaporation.

### 2.4. Crystallography

Single crystals of the compounds were obtained from the reaction mixture according to the described synthetic procedure after slow evaporation. For the structural determination of each compound, one well-shaped single crystal of the respective compound was mounted on a Bruker Kappa APEX II diffractometer equipped with a triumph monochromator at ambient temperature. Diffraction measurements were recorded using Mo *Kα* radiation. Unit cell dimensions were determined using at least 227 reflections in the range $10 < \theta < 20°$. Intensity data were collected using $\varphi$ and $\omega$ scan mode. The frames collected for the crystal were integrated with the Bruker SAINT software package [22] using a narrow-frame algorithm. Data were corrected for absorption using the numerical method (SADABS) based on crystal dimensions [23]. The structure was solved using the SUPERFLIP [24] package and refined by full-matrix least-squares method on $F^2$ using the CRYSTALS package version 14.61build6236 [25]. All non-disordered, non-hydrogen atoms have been refined anisotropically. In the case of the silver complex, one nitrate anion was found disordered over two positions with equal occupation factors of one half for each. One half of a water solvent molecule per asymmetric unit was also found disordered. All non-hydrogen atoms of these counter anions/solvent molecules have been isotropically refined with fixed occupation factors.

Hydrogen atoms riding on non disordered parent atoms were located from difference Fourier maps and refined at idealized positions riding on the parent atoms with isotropic displacement parameters Uiso(H) = 1.2Ueq(C) or 1.5Ueq(-OH hydrogens) at distances C-H 0.95 Å and O-H 0.82 Å. All OH hydrogen atoms were allowed to rotate, but not to tip. Hydrogen atoms riding on disordered oxygen atoms of water solvent molecules were positioned geometrically to fulfill hydrogen bonding demands. The crystal data, details of data collection, and structure refinement for the compound studied are given in Table 1. Illustrations were drawn by CAMERON [26] and Diamond [27]. Further details on the crystallographic studies, as well as atomic displacement parameters, are given as Supporting Information in the form of .cif files (Supplementary Material).

**Table 1.** Crystal data and experimental details for the studied compounds.

| Crystal Data | | |
|---|---|---|
| Chemical formula<br>Moiety formula | $C_{90}H_{72}B_2Cu_2F_8N_8P_4$<br>$C_{90}H_{72}Cu_2N_8P_4$, $2(BF_4)$ | $C_{90}H_{75}Ag_2N_{10}O_{7.50}P_4$<br>$C_{90}H_{72}Ag_2N_8P_4$, $2(NO_3)$,<br>$1.5(H_2O)$ |
| $M_r$ | 1690.21 | 1756.28 |
| Crystal system,<br>Space group | Monoclinic<br>$P2_1/n$ | Triclinic<br>$P\bar{1}$ |
| Temperature (K) | 295 | 295 |
| $a$ (Å)<br>$b$ (Å)<br>$c$ (Å)<br>$\alpha$ (°) | 11.4136 (12)<br>25.281 (3)<br>14.3061 (17)<br>90 | 15.771 (2)<br>17.242 (2)<br>19.635 (3)<br>64.301 (4) |
| $\beta$ (°)<br>$\gamma$ (°) | 104.919 (4)<br>90 | 68.549 (4)<br>62.873 (4) |
| $V$ (Å³) | 3988.8 (8) | 4186.8 (10) |
| $Z$ | 2 | 2 |
| Radiation type | Mo *Kα* | Mo *Kα* |
| $\mu$ (mm⁻¹) | 0.69 | 0.61 |
| Crystal size (mm) | $0.14 \times 0.13 \times 0.11$ | $0.20 \times 0.19 \times 0.12$ |
| $T_{min}$, $T_{max}$ | 0.91, 0.93 | 0.89, 0.93 |

**Table 1.** *Cont.*

| No. of reflections | | |
| --- | --- | --- |
| Measured | 35,566 | 70,037 |
| Independent | 7633 | 16,094 |
| Observed [$I > 2.0\sigma(I)$] | 5673 | 11,509 |
| $R_{int}$ | 0.055 | 0.029 |
| $(\sin \theta / \lambda)_{max}$ (Å$^{-1}$) | 0.612 | 0.615 |
| Refinement | | |
| $R[F^2 > 2\sigma(F^2)]$, | 0.045 | 0.046 |
| $wR(F^2)$, | 0.096 | 0.108 |
| $S$ | 1.00 | 1.00 |
| No. of reflections | 5673 | 11,509 |
| No. of parameters | 514 | 1018 |
| No. of restraints | - | 8 |
| $\Delta\rho_{max}$, $\Delta\rho_{min}$ (e Å$^{-3}$) | 0.57–0.39 | 1.27–0.74 |

## 3. Results and Discussion

The characteristic imine band in the IR spectrum of the ligand was identified at 1654 cm$^{-1}$. Its shift to 1605–1612 cm$^{-1}$ in the products helps to establish the coordination of the diimine site to the metal ion. Additional evidence was provided by the presence of the characteristic bands of the tetrafluoroborate and nitrate ions at 1054 and 1370 cm$^{-1}$, respectively. The C–P bands of the phosphines were observed in the region 506–534 cm$^{-1}$.

The single proton ortho to a heterocyclic N atom (position 6 at the pyridine ring) was observed at 8.92 ppm in the $^1$H NMR of the ligand, while the H-3 of the same ring appeared at 8.71 ppm. The rest of the protons lied in the region 7.36–7.94 ppm. The imino carbon atom appeared at 160.7 ppm in the $^{13}$C NMR spectrum. No attempt was made to further analyze the aromatic region of the NMR spectra, due to the proximity to the values reported for the ligand by its manufacturers.

### 3.1. Crystal Structure of [{Cu(L)}$_2$(μ-dppm)$_2$](BF$_4$)$_2$

The compound crystallizes in the monoclinic $P2_1/n$ space group and the unit cell comprised two binuclear bicationic copper complexes and four BF$_4$ counter anions. The molecular structure of the compound is given in Figure 1; selected bond distances and angles are listed in Table 2. The asymmetric unit contained one copper cation, one diphosphine dppm ligand, one diimine ligand, and one tetrafluoroborate counter anion. The final binuclear complex was formed when the asymmetric unit duplicates to the local intramolecular inversion center found on the mid distance Cu(I)-Cu(II). Each binuclear complex was formed as the four phosphorous atoms from two dppm ligands bridge two Cu(I) metal centers. The diphosphine presents a highly-strained, four-membered heterocyclic ring, if it were to adopt chelating coordination to the metal, and so its bridging coordination was expected. Each copper cation was coordinated to two phosphorous atoms and was also coordinated to two nitrogen atoms from a heterocyclic diimine ligand L. This gave rise to an eight-membered ring with the participation of two metal centers, two carbon atoms, and four phosphorus atoms from the two bridging diphenylphosphinemethane. The heterocyclic diimine ligand was coordinated with the pyridine nitrogen cis to the N–N site of the adjacent ring. The π systems of two phenyl rings of the same biphosphine ligand adopted π–π stacking in the eclipsed conformation, with a centroid-to-centroid distance of 3.553 Å, while the minimum C–C distance was 3.287 Å. Another π–π interaction was found, as one more phosphine phenyl ring interacted with the triazolium ring of the diimine, with centroid to centroid distance of 3.904 Å and a minimum C(40)-N(2) distance of 3.292 Å. The coordination of the biphosphine was slightly asymmetric, giving rise to Cu-P distances of

2.2259 Å and 2.2610 Å. The diimine coordination appeared less asymmetric, with Cu-N distances of 2.113 Å and 2.122 Å. The longer Cu-N distance was observed towards the pyridine ring nitrogen atom. The coordination environment of the metals was distorted tetrahedral, with the larger angle (130.85°) being the one of the interligand P-Cu-P and the smaller one (78.40°) observed for the diimine coordination.

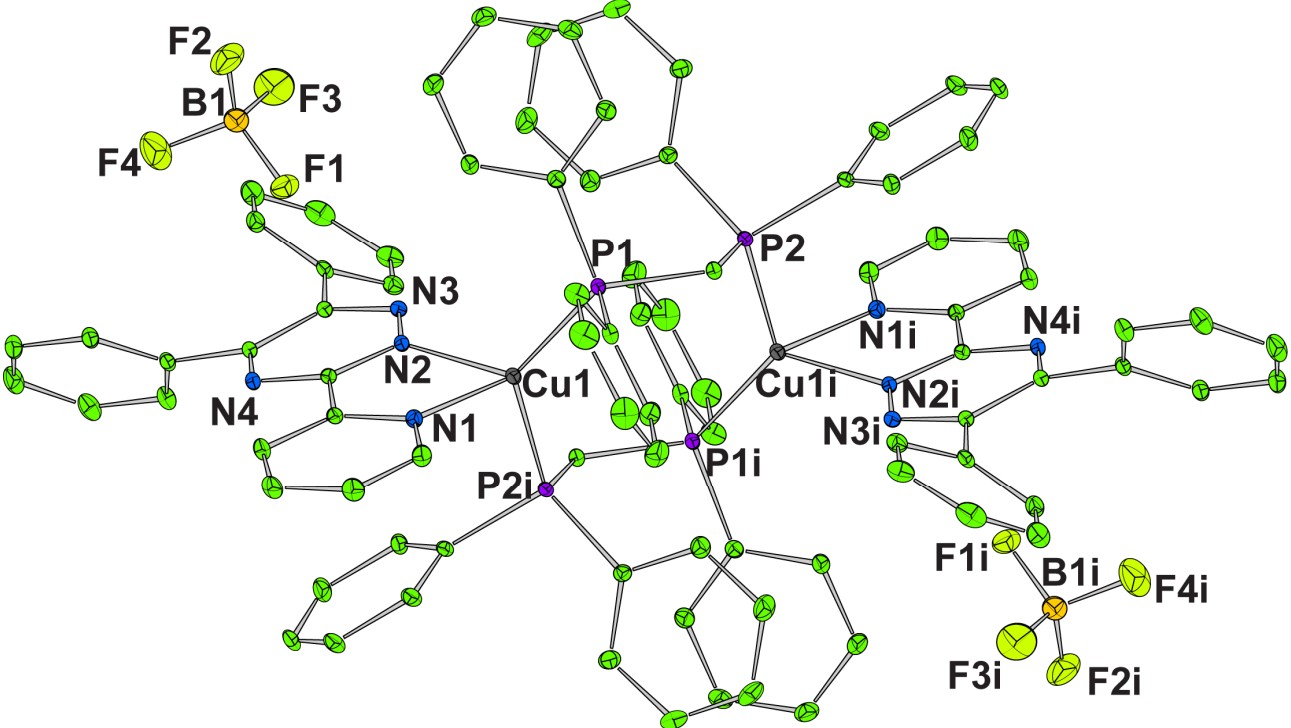

**Figure 1.** Figure of $C_{90}H_{72}Cu_2N_8P_4$, 2(BF$_4$). Hydrogen atoms have been omitted. Ellipsoids with 30% probability.

**Table 2.** Selected geometric parameters (Å, °) for $C_{90}H_{72}Cu_2N_8P_4$, 2(BF$_4$).

| Cu1—P2$^i$ | 2.2610 (8) | P2$^i$—Cu1—P1 | 130.85 (3) |
|---|---|---|---|
| Cu1—P1 | 2.2259 (8) | P2$^i$—Cu1—N1 | 106.49 (7) |
| Cu1—N1 | 2.122 (3) | P1—Cu1—N1 | 107.45 (7) |
| Cu1—N2 | 2.113 (2) | P2$^i$—Cu1—N2 | 93.42 (7) |

The coordination behavior of the ligand resembles the one observed in a Cu(II) compound, namely [CuL$_2$(H$_2$O)$_2$][ClO$_4$]$_2$, where the two Cu-N distances are very close to each other (2.036 Å and 2.033 Å), the N-Cu-N angle is 79.7, and the two phenyl rings on the triazine ring form dihedral angles of 35.2° and 52.0°, respectively [28]. The phenyl rings adopt a geometric displacement similar to the one observed in the free ligand where the corresponding torsion angles are 51.09° and 33.36° [29].

### 3.2. Crystal Structure of [{Ag(L)}$_2$(μ-dppm)$_2$] (NO$_3$)$_2$ 1.5(H$_2$O)

The compound crystallizes in the triclinic *P*ī space group and the unit cell comprised two bicationic binuclear silver complexes, totally four partially disordered nitrate counter ions and totally three partially disordered solvate water molecules. Since there was no purification of the reagents used or processes to eliminate moisture during the syntheses, the most probable origin of the water molecules is isopropanol. During the synthesis of the ligand, some water molecules could have been captured through hydrogen bonds and then transferred to the final product. The molecular structure of the compound is given in Figure 2; selected bond distances and angles are listed in Table 3. The asymmetric unit presented no local intramolecular inversion center and comprised the whole bicationic

binuclear complex, as well as totally two nitrate counter anions and one and a half disordered water solvate molecules. The structure of the binuclear silver complex (even not centrosymmetric) was similar to the one of the Cu(I) discussed above. The coordination of the diphosphine was asymmetric with Ag–P distances of 2.4168 Å and 2.4544 Å. Diimine coordination was even more asymmetric, giving rise to Ag-N bond lengths of 2.463 Å and 2.598 Å, with the smaller one corresponding to the nitrogen of the pyridine ring. Due to the larger ionic radius of silver, relative to copper, the compound revealed an even larger distortion than its copper analogue around the metal, with the P–Ag–P angle being equal to 159.91° and revealing a great similarity to seesaw coordination geometry. Stacking interactions were, again, present between the phenyl rings of the same phosphine. In this case, all adopted the parallel conformation and could be found between phenyl rings of both phosphines with centroid to centroid distances of 3.621 and 3.675 Å. Interactions of π–π type were also found between phenyl rings of the phosphines and the pyridyl rings of the diimino ligands, with centroid to centroid distances of 3.747 Å and 4.110 Å, with a minimum distance of 3.407 Å between N(5) and C(85) and 3.553 Å between N(1) and C(66). Hydrogen bonding interactions kept the nitrate counter ions and the solvate water molecules close, giving stability to the crystal lattice.

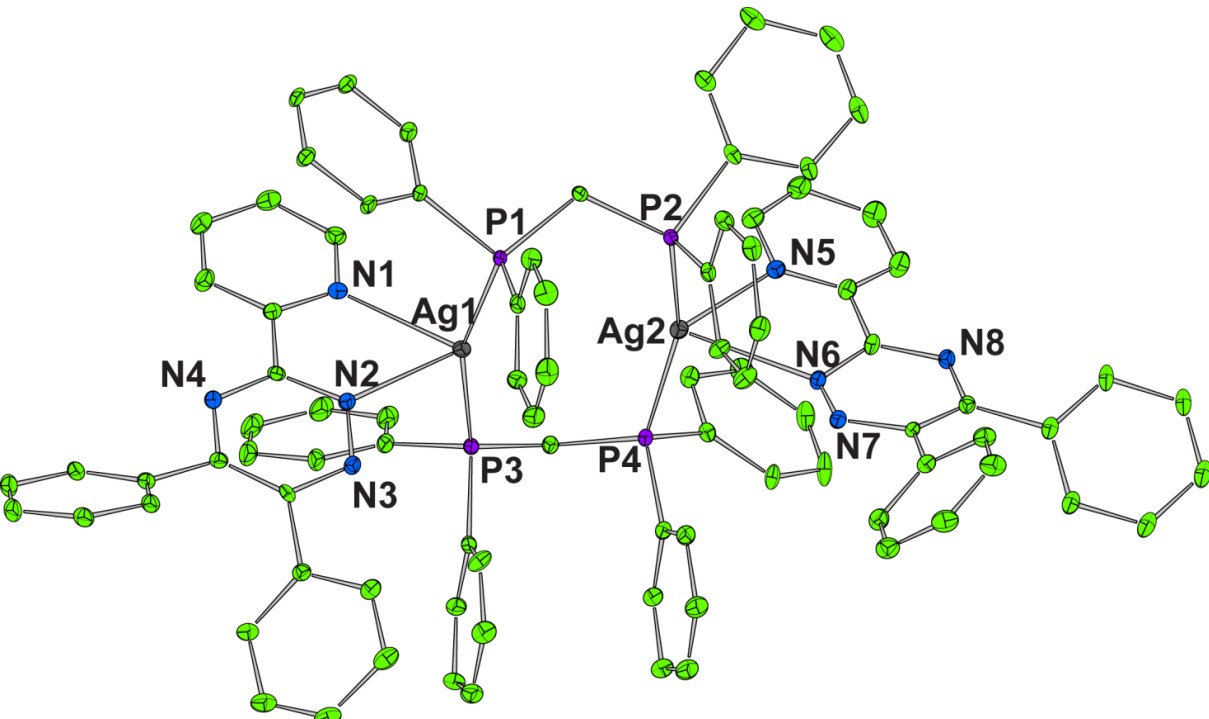

**Figure 2.** Figure of $C_{90}H_{72}Ag_2N_8P_4$, 2(NO$_3$), 1.5(H$_2$O). Hydrogen atoms, counter ions, and solvent molecules have been omitted. Ellipsoids with 30% probability.

**Table 3.** Selected geometric parameters (Å, °) for $C_{90}H_{72}Ag_2N_8P_4$, 2(NO$_3$), 1.5(H$_2$O).

| | | | |
|---|---|---|---|
| Ag1—P1 | 2.4526 (10) | P1—Ag1—P3 | 159.91 (4) |
| Ag1—P3 | 2.4168 (10) | P1—Ag1—N1 | 98.05 (8) |
| Ag1—N1 | 2.463 (4) | P3—Ag1—N1 | 100.75 (8) |
| Ag1—N2 | 2.565 (3) | P1—Ag1—N2 | 94.86 (8) |
| Ag2—P2 | 2.4197 (10) | P3—Ag1—N2 | 99.43 (8) |
| Ag2—P4 | 2.4544 (10) | N1—Ag1—N2 | 64.99 (11) |
| Ag2—N5 | 2.491 (4) | P2—Ag2—P4 | 159.83 (4) |
| Ag2—N6 | 2.598 (3) | P2—Ag2—N5 | 101.69 (9) |

The diphenylphosphinemethane molecule acted as a bridge and not as a chelating agent, since in the latter coordination, a rather strained four-member ring would emerge. The relative flexibility of the core eight-member ring formed by the two compounds was expected to be labile and, upon excitation, could reach a state from which relaxation would probably occur in a radiation less fashion. Semiempirical quantum chemical calculations, using the MOPAC program [30], predicted that the ligand, in its chelating form, presents a bipyridine-like skeleton, for which the HOMO-LUMO gap was substantially smaller than the corresponding one in bipyridine. This finding provides grounds for the assumption that, in the actual complexes, the mainly intraligand or metal-to-ligand-charge transfer excitations will be to states lying lower in energy to the emitting ones, relative to bipyridine analogues. In this case, the radiation less relaxation would give rise to emission at higher wavelengths than the analogous bipyridine compounds, or none at all. The energies and constitution of the frontier orbitals of the ligand studied and of bipyridine are shown in Figure 3.

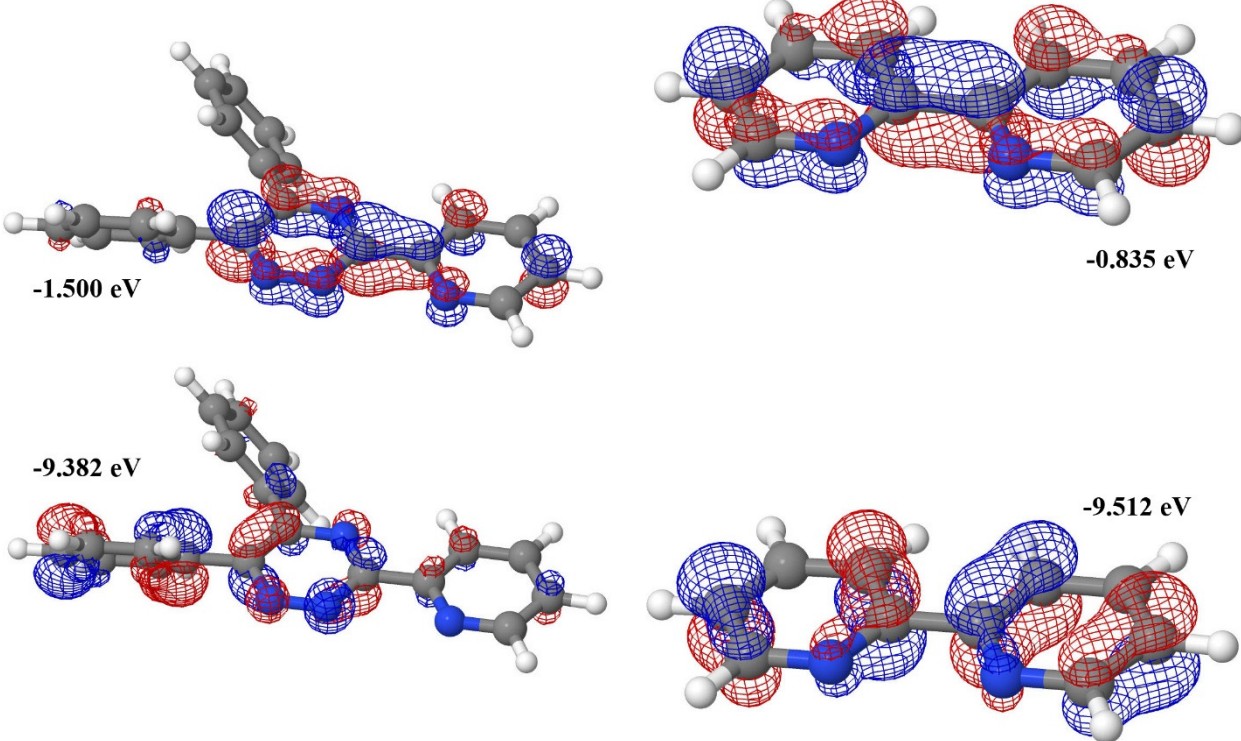

**Figure 3.** Frontier molecular orbital energies calculated by MOPAC for the ligand used (**left**) and for bipiridine (**right**).

Diffuse reflectance spectra were recorded for the compounds, which were dominated by strong absorption bands below 500 nm, where mainly charge transfer electron excitations were expected to occur (Figure 4, left side). The copper compound revealed a larger red shift than the silver one. The excitation spectrum of the copper compound was identical to the diffuse reflectance spectrum, and its low intensity emission was observed at about 250 nm higher than the excitation maximum (Figure 4, right side).

The free rotation of the phenyl rings on the diimine backbone is the most probable reason for the low intensity emission in the copper compound. In confirmation of the above, the deuteration of bipyridine ligand and excited state lifetime observations of [Ru(bpy)$_2$L](PF$_4$)$_2$ confirmed that the emitting state of the complex was located on the triazine ligand [31]. The kinetic liability of silver complexes, and especially their ability to undergo ligand scrambling reactions, is well-documented [32,33]; therefore, the complete absence of emission in the silver complex was not unexpected.

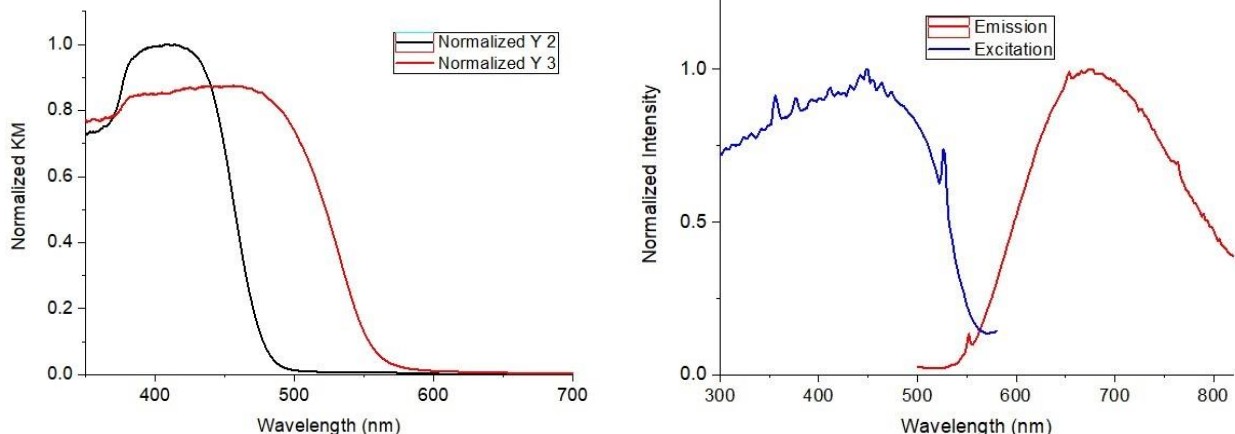

**Figure 4.** On the left, diffuse reflectance spectra of the Cu (red) and Ag (black) compounds. On the right, excitation and emission spectra of the Cu compound. Both spectra have been normalized.

## 4. Conclusions

The diimine site of 5,6-Diphenyl-3-(2-pyridyl)-1,2,4-triazine coordinates readily to Cu(I) and Ag(I) and forms dimeric compounds in the presence of bis(diphenylphosphine) methane, which acts as a bridge between the metal centres. The spectroscopic data of the complexes reveal only slight structural modifications of the diimine ligand upon coordination. The solid-state visible spectra of the compounds are dominated by the charge transfer electron excitation bands, while only the copper complex reveals weak emission. This observation can be attributed to the possibility of the substituent phenyl rings' rotation and to the HOMO-LUMO gap, in which semiempirical calculations proposed that is smaller than the one realized for analogous 2,2′-bipyridine compounds.

**Supplementary Materials:** The following supporting information can be downloaded at: https: //www.mdpi.com/article/10.3390/chemistry5030103/s1, Supplementary Crystallographic Data: CCDC 2235299 and CCDC 2235300 contain the supplementary crystallographic data for this paper. These data can be obtained free of charge via www.ccdc.cam.ac.uk/conts/retrieving.html (or from the Cambridge Crystallographic Data Centre, 12 Union Road, Cambridge CB21EZ, UK; fax: (+44) 1223-336-033; or deposit@ccde.cam.ac.uk).

**Author Contributions:** Conceptualization, D.T. and P.D.A.; methodology, D.T. and A.S.; software, A.H. and P.D.A.; validation, A.H. and D.T.; formal analysis, A.S.; investigation, A.S. and P.D.A.; resources, A.H. and D.T.; data curation, A.H.; writing—original draft preparation, D.T.; writing—review and editing, D.T.; visualization, A.H.; supervision, D.T.; project administration, P.D.A. All authors have read and agreed to the published version of the manuscript.

**Funding:** This research received no external funding.

**Conflicts of Interest:** The authors declare no conflict of interest.

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
