# Peer review of "Synthesis of the Bipyridine-Type Ligand 3-(2-Pyridyl)-5,6-diphenyl-1,2,4-triazine and Structural Elucidation of Its Cu(I) and Ag(I) Complexes"

_chemistry, doi:10.3390/chemistry5030103_

Round 1

Reviewer 1 Report

The paper of A. Harzidimitriou and co-authors is a fundamental work on synthesis two new Copper(I) and Ag(I) complexes with substituted diimine ligand. Authors obtained compounds in a single crystal form, determined structure with X-RAY analysis. The resulting compounds are dimers with diphosphine bridges. The article is a classic "crystal-structure-article" case. Enough scientific data to publish. However, a very large revision of the article is needed.

Major:

The first and most important remark is that it is not clear that the authors did not synthesize the ligand for the first time. This is not stated in the manuscript, but there are no references to earlier published papers. This is not a new ligand, and it is unethical to talk about its synthesis in the title, in the abstract, to describe its synthesis in an article without mentioning the source.

https://scripts.iucr.org/cgi-bin/paper?S1600536807004229 – here is Single Crystal Structure, 2007

The second unclear point is whether the structures are deposited with the Cambridge Structural Database or not. I didn't find it in the text of the article. Deposit structures and give their numbers.

2.3 - It is necessary to describe the experimental part in more detail: describe the yield of products, IR spectra with vibrations, NMR spectra with peaks

“One half of a methanol solvent molecule per asymmetric unit was also found disordered” - if I understand correctly, the reaction was in acetonitrile and methylene chloride. Crystals were obtained from diethyl ether solution. Where did methanol come from? Where did the water molecule come from in the case of the silver complex?

“Totally one acetonitrile solvent molecule per complex unit was found disordered” - As far as I can see, this information is not displayed in Table 1. Is the acetonitrile molecule included in the chemical formula?

“Supporting Information in the form of cif files” – absent in SuSy

There are no conclusions in the article

Minor:

It would be better to add the chemical scheme of the reaction

Use proper degree sign ° instead of â–«/ο/â—¦/capture to Table 1/3

And typos:

whch, allowes, wash (washed?), shapedsingle, compoundwas, atomsof, Figrue 3

Author Response

The paper of A. Harzidimitriou and co-authors is a fundamental work on synthesis two new Copper(I) and Ag(I) complexes with substituted diimine ligand. Authors obtained compounds in a single crystal form, determined structure with X-RAY analysis. The resulting compounds are dimers with diphosphine bridges. The article is a classic "crystal-structure-article" case. Enough scientific data to publish. However, a very large revision of the article is needed.

 Major:

The first and most important remark is that it is not clear that the authors did not synthesize the ligand for the first time. This is not stated in the manuscript, but there are no references to earlier published papers. This is not a new ligand, and it is unethical to talk about its synthesis in the title, in the abstract, to describe its synthesis in an article without mentioning the source.

Now both the initial (1981) and the most recent (2003) variation of the ligand synthesis are reported and is it definitely stated in the text that the described is a locally modified synthetic process for the ligand, which for many years is commercially available.

https://scripts.iucr.org/cgi-bin/paper?S1600536807004229 – here is Single Crystal Structure, 2007

In the experimental part of this manuscript it is stated that the compound was purchased from a commercial manufacturer.

The second unclear point is whether the structures are deposited with the Cambridge Structural Database or not. I didn't find it in the text of the article. Deposit structures and give their numbers.

 We do apologize for the dramatic misunderstanding with the crystallographer was misinformed about which structures were required as we were currently dealing with multiple manuscripts. The appropriate cif files uploaded and deposition numbers reported in the text in the supplementary information section.

2.3 - It is necessary to describe the experimental part in more detail: describe the yield of products, IR spectra with vibrations, NMR spectra with peaks

 The relevant IR bands are reported and the main NMR features described. However, as we note in the text their match with the reported values for the commercially available ligand do not make it mandatory, in our belief, to replicate the information.

“One half of a methanol solvent molecule per asymmetric unit was also found disordered” - if I understand correctly, the reaction was in acetonitrile and methylene chloride. Crystals were obtained from diethyl ether solution. Where did methanol come from? Where did the water molecule come from in the case of the silver complex?

 in the Experimental where we state that there was no purification procedure for the chemicals purchased, therefore the solvents may contain minute percentage of water which is incorporated into the crystal lattice. Methanol was obviously a typing error, referring to some structure that we were working on at the moment.

“Totally one acetonitrile solvent molecule per complex unit was found disordered” - As far as I can see, this information is not displayed in Table 1. Is the acetonitrile molecule included in the chemical formula?

Thank you for the remark. There was a typing error which has been corrected accordingly

“Supporting Information in the form of cif files” – absent in SuSy

 The comment with the wrong inquiry with the crystallographer described above, is the source of this missing information. Now the deposition of the data has been completed and the relevant information is included in the text.

There are no conclusions in the article

 A short conclusion section has been added.

Minor:

It would be better to add the chemical scheme of the reaction

In our opinion it would have to be two schemes or one with several generalizations (i.e. M- Cu or Ag, etc). In any case the only interesting part would be the structure of the final product which is depicted in the crystal structure schemes.

Use proper degree sign ° instead of â–«/ο/â—¦/capture to Table 1/3

And typos:

whch, allowes, wash (washed?), shapedsingle, compoundwas, atomsof, Figrue 3

All those that came into our notice have been taken care of.

Reviewer 2 Report

In this article, the authors synthesize two metal complexes based on diimine ligands. These complexes have been characterized by 1H and 13C spectroscopy but preferably by X-ray spectroscopy. There are some comments on the article which are detailed below:

Introduction

In general, the information provided indicates the importance of this topic, however it should be revised, as some of the sentences are unconnected and present meaningless information due to the lack of verbs in them.

Furthermore,

In line 3, Ag+ and Cu+ are indicated while Ag (I) and Cu (I) are used in the rest of the article.

Line 4 states "while silver diphosphine" and should read "while gold diphosphine ?".

In line 11, it says "long lived MLCT excited states", MLCT stands for metal to ligand charge transfer, but it is not cited in the text, so perhaps there should be a reference the first time it appears.

Experimental

2.2 synthesis of the diimine ligand

I don't know if the expression "the solution was brought" is correct, but in my opinion "the solution was cooled" could be better because it changes from rt to 0ºC.

Synthesis of the complexes

"diphosphine dppm" should be indicated at least the first time as 1,1, bis-(diphenylphosphine)methane. Check the spelling in this paragraph as some of the words are not spelled correctly.

Result and discussion

It is not clear from the first sentence of the discussion whether the imine crystals show 4 Cu and 4 BF4 or only two, as the subsequent discussion, as well as the table and drawing, indicate only 2. The same could be said for the other complex.

Conclusion

Some sentences were missing as a conclusion to what was stated in the work.

References

Diimine ligands are a very topical subject, a reference from 1982, although it could be correct, is very old. There are more recent references related to this topic.

Ref 12, there are two references here, they should be separated as a and b.

Ref 14 does not fit in the references section, the order of authors and title is missing.

Some references indicated the doi of the article and many did not.

Author Response

In this article, the authors synthesize two metal complexes based on diimine ligands. These complexes have been characterized by 1H and 13C spectroscopy but preferably by X-ray spectroscopy. There are some comments on the article which are detailed below:

Introduction

In general, the information provided indicates the importance of this topic, however it should be revised, as some of the sentences are unconnected and present meaningless information due to the lack of verbs in them.

Furthermore,

In line 3, Ag+ and Cu+ are indicated while Ag (I) and Cu (I) are used in the rest of the article.

Uniform presentation of the oxidation states of the metals is present, in the form of Ag(I) as Ag+ has been dropped.

Line 4 states "while silver diphosphine" and should read "while gold diphosphine ?".

It is silver actually

In line 11, it says "long lived MLCT excited states", MLCT stands for metal to ligand charge transfer, but it is not cited in the text, so perhaps there should be a reference the first time it appears.

Explanation of the few abbreviations used is given at the first appearance of each one in the text.

Experimental

2.2 synthesis of the diimine ligand

I don't know if the expression "the solution was brought" is correct, but in my opinion "the solution was cooled" could be better because it changes from rt to 0ºC.

Sentence corrected

Synthesis of the complexes

"diphosphinedppm" should be indicated at least the first time as 1,1, bis-(diphenylphosphine)methane. Check the spelling in this paragraph as some of the words are not spelled correctly.

Explanation of the few abbreviations used is given at the first appearance of each one in the text. Spelling should also be all right now.

Result and discussion

It is not clear from the first sentence of the discussion whether the imine crystals show 4 Cu and 4 BF4 or only two, as the subsequent discussion, as well as the table and drawing, indicate only 2. The same could be said for the other complex.

In the unit cell of both crystals two bicationic binuclear complexes can be found together with four counter ions (BF4 or NO3 depending on the complex). Chemical formula sum and moiety formula in Table 1 combined with Z gives the ingredients of the unit cell in each case

Conclusion

Some sentences were missing as a conclusion to what was stated in the work.

A short conclusion section has been added to the text

References

Diimine ligands are a very topical subject, a reference from 1982, although it could be correct, is very old. There are more recent references related to this topic.

Ref 12, there are two references here, they should be separated as a and b.

Ref 14 does not fit in the references section, the order of authors and title is missing.

Some references indicated the doi of the article and many did not.

DOIs were found to be not mandatory so they were dropped some time during the preparation of the manuscript. They are omitted altogether now.

References 12 and 14 have been amended according to the indications.

Both the initial (1981) and the most recent (2003) variation of the ligand synthesis are reported at a reviewer’s request. In this respect a reference to some of the pioneering works on diimines may be regarded as being historically correct rather than revealing ignorance of the current literature.

By adding the two references for the above point the numbering of the references has altered from 19 onwards.

Round 2

Reviewer 1 Report

Dear Authors, Editor,

The manuscript takes into account almost all my recommendations, in my opinion it has become better, some points have become clearer. In this form, I recommend accepting it for publication in the Chemistry.

Good luck!